# Ergotamine Stimulates Human 5-HT_4_-Serotonin Receptors and Human H_2_-Histamine Receptors in the Heart

**DOI:** 10.3390/ijms24054749

**Published:** 2023-03-01

**Authors:** Hannes Jacob, Pauline Braekow, Rebecca Schwarz, Christian Höhm, Uwe Kirchhefer, Britt Hofmann, Joachim Neumann, Ulrich Gergs

**Affiliations:** 1Institute for Pharmacology and Toxicology, Medical Faculty, Martin Luther University Halle-Wittenberg, 06120 Halle (Saale), Germany; 2Institute for Pharmacology and Toxicology, Medical Faculty, Westfälische Wilhelms-Universität, 48149 Münster, Germany; 3Department of Cardiac Surgery, Mid-German Heart Center, University Hospital Halle, 06120 Halle (Saale), Germany

**Keywords:** ergotamine, human atrium, mouse atrium, mouse ventricle

## Abstract

Ergotamine (2′-methyl-5′α-benzyl-12′-hydroxy-3′,6′,18-trioxoergotaman) is a tryptamine-related alkaloid from the fungus *Claviceps purpurea*. Ergotamine is used to treat migraine. Ergotamine can bind to and activate several types of 5-HT_1_-serotonin receptors. Based on the structural formula of ergotamine, we hypothesized that ergotamine might stimulate 5-HT_4_-serotonin receptors or H_2_-histamine receptors in the human heart. We observed that ergotamine exerted concentration- and time-dependent positive inotropic effects in isolated left atrial preparations in H_2_-TG (mouse which exhibits cardiac-specific overexpression of the human H_2_-histamine receptor). Similarly, ergotamine increased force of contraction in left atrial preparations from 5-HT_4_-TG (mouse which exhibits cardiac-specific overexpression of the human 5-HT_4_-serotonin receptor). An amount of 10 µM ergotamine increased the left ventricular force of contraction in isolated retrogradely perfused spontaneously beating heart preparations of both 5-HT_4_-TG and H_2_-TG. In the presence of the phosphodiesterase inhibitor cilostamide (1 µM), ergotamine 10 µM exerted positive inotropic effects in isolated electrically stimulated human right atrial preparations, obtained during cardiac surgery, that were attenuated by 10 µM of the H_2_-histamine receptor antagonist cimetidine, but not by 10 µM of the 5-HT_4_-serotonin receptor antagonist tropisetron. These data suggest that ergotamine is in principle an agonist at human 5-HT_4_-serotonin receptors as well at human H_2_-histamine receptors. Ergotamine acts as an agonist on H_2_-histamine receptors in the human atrium.

## 1. Introduction

Ergotamine (Figure 1B), tested in this work, is currently mainly used in the clinic for the treatment of migraine [1,2]. Ergotamine can bind to and activate 5-HT_1_-receptors and 5-HT_2A_-receptors in the brain [2]. Ergotamine can lead to hallucinations probably via these 5-HT_2A_-receptors [3,4]. Ergotamine can cause vasoconstriction probably because ergotamine stimulates peripheral vascular 5-HT_2A_-receptors and peripheral vascular α_1_-adrenoceptors [4]. Ergotamine is found in fungi like *Claviceps pururea* (Secale cornutum). Secale cornutum can be found in cereals (e.g., rye grain) and causes arterial constrictions (via stimulation of α-adrenoceptors), but possibly also hallucinations (e.g., [5,6,7,8]).

Ergotamine is formed in fungi from lysergic acid, to which, alanine, proline and phenylalanine are covalently linked [9]. No inotropic effect of ergotamine was found in isolated paced cat papillary muscle [10]. However, that might be a species problem, as H_2_-histamine- and 5-HT_4_-receptors are functionally absent in the cat heart [11,12,13]. A close derivative of ergotamine, called ergometrine (lysergic acid β-propanolamide), has, in contrast, been shown to elicit an increase in force in the guinea pig heart [14], which contains functional H_2_-histamine receptors [13]. The increase in force of contraction caused by ergometrine was antagonized by cimetidine, which convincingly suggested a H_2_-histamine receptor-mediated effect of ergometrine in the guinea pig heart [14]. Similarly, lysergic acid diethylamide (LSD) increased force of contraction in isolated guinea pig and rabbit cardiac preparations [15]. Hence, ergotamine might increase force of contraction in those species that contain functionally active H_2_-receptors in the heart. To the best of our knowledge, a positive inotropic cardiac effect by ergotamine in any mammalian species has never been published before. As ergotamine binds to some isoforms of peripheral serotonin-receptors [16], we hypothesized that ergotamine might stimulate human serotonin receptors in the heart. In the human heart, all inotropic and chronotropic effects of serotonin are mediated via 5-HT_4_-receptors [17,18]. These 5-HT_4_-receptors are lacking, in a functional manner, in mouse heart: serotonin failed to alter the force of contraction in isolated mouse cardiac preparations from wild type mice (WT, [19,20,21,22,23]). To facilitate the study of human 5-HT_4_-receptors, we had previously produced and characterized intensively transgenic mice (5-HT_4_-TG) with cardiac expression of this receptor only in the heart, which responded with an increase in force and frequency to serotonin [14,19,21,22]. Therefore, we decided to test the hypothesis that ergotamine would exert positive inotropic and positive chronotropic effects in these 5-HT_4_-TG (Figure 1A).

In the heart, four histamine receptor subtypes have been found [13]. However, species differences, regional differences and cellular differences in histamine receptor function exist in the heart [13]. In the mouse heart, histamine can only release noradrenaline [22,23,24,25,26,27,28]. Similar to the mouse, the rat, dog and cat experienced the effects of histamine resulting from a release of endogenous catecholamines [11,15,29,30,31,32].

In humans, H_2_-histamine receptors are present in both the atrium and ventricle (radioligand binding: [33], antibody and RNA expression: [34]). The cardiac H_2_-histamine receptors mediate the positive inotropic effects of histamine in isolated human atrial cardiac preparations [35,36,37]. Therefore, we have generated transgenic mice that overexpress the H_2_-histamine receptors only in the heart (H_2_-TG), wherein histamine increases force of contraction [22,23,24,25,26,27,28].

Hence, we tested the following hypotheses: ergotamine might increase force of contraction in 5-HT_4_-TG and/or in H_2_-TG, and in human atrial preparations via HT_4_- and/or H_2_-receptors. Progress reports have been published in abstract form [38].

## 2. Results

### 2.1. H_2_-TG Atrium

#### 2.1.1. Left Atrium

We noticed that ergotamine time- and concentration-dependently increased force of contraction in H_2_-TG. A typical original recording is seen in Figure 2A. For comparison, we studied WT. In WT, ergotamine failed to increase force of contraction (Figure 2B). In H_2_-TG additionally applied histamine, force of contraction further failed to increase (Figure 2A), while additionally applied histamine was ineffective in left atrium from WT (Figure 2B). Summarizing the results, one can see that ergotamine concentration-dependently increased force of contraction in left atrial preparations (Figure 2C). Moreover, ergotamine concentration-dependently shortened the time to peak tension (Figure 2D). This shortening was so extensive that additionally applied histamine could not shorten time to peak tension any further. In a similar fashion, ergotamine hastened time of relaxation concentration-dependently (Figure 2E) and additionally applied histamine was not more effective to shorten time of relaxation (Figure 2E). In addition, ergotamine also concentration dependently enhanced the absolute value of the rate-of-tension development (Figure 2F) and the rate-of-tension relaxation, and this effect of ergotamine was maximal because additionally applied histamine augmented these parameters no further. Hence, ergotamine acted as a full agonist at H_2_-histamine receptors under these conditions (Figure 2G).

#### 2.1.2. Right Atrium

In right atrial preparations from H_2_-TG, ergotamine increased the beating rate as seen in an original recording (Figure 3A). Additionally applied histamine did not increase the beating rate any further (Figure 3A). Ergotamine failed to increase the beating rate in WT. Several such experiments are summarized in Figure 3B.

### 2.2. 5-HT_4_-TG

#### 2.2.1. Left Atrium

We noticed that ergotamine time- and concentration-dependently increased force of contraction in 5-HT_4_-TG left atria. A typical original recording is seen in Figure 4A. For comparison, we studied WT: here, ergotamine failed to increase force of contraction (Figure 4B). In 5-HT_4_-TG, additionally applied 5-HT increased force of contraction further (Figure 2A), while ergotamine and 5-HT were ineffective in left atrium from WT (Figure 4B). Summarizing the results, one can see that ergotamine concentration-dependently increased force of contraction in left atrial preparations from 5-HT_4_-TG (Figure 4C). Moreover, ergotamine concentration-dependently shortened the time of tension in 5-HT_4_-TG (Figure 4D). This shortening was so extensive that additionally applied serotonin could not shorten time to peak tension any further. In a similar fashion, ergotamine hastened time of relaxation concentration-dependently (Figure 4E) and additionally applied histamine was not more effective to reduce time of relaxation (Figure 4E) in 5-HT_4_-TG. In addition, ergotamine also concentration-dependently enhanced the absolute value of the rate-of-tension development (Figure 4F) and the rate-of-tension relaxation (Figure 4G) in 5-HT_4_-TG.

#### 2.2.2. Right Atrium

In right atrial preparations from 5-HT_4_-TG, ergotamine increased the beating rate as seen in an original recording (Figure 5A). Additionally applied serotonin hardly increased the beating rate any further in 5-HT_4_-TG (Figure 5A). Ergotamine failed to increase the beating rate in WT. Several such experiments are summarized in Figure 5B: Ergotamine concentration-dependently increased the beating rate in isolated right atrial preparations (Figure 5B). This effect was also antagonized by subsequently applied tropisetron.

### 2.3. Isolated Perfused Hearts

It was interesting to study whether ergotamine might affect left ventricular function, because in the mouse and human heart, the left ventricle is decisive for the perfusion of the organs of the body. Therefore, we perfused retrogradely isolated spontaneously beating whole hearts (Langendorff procedure) from H_2_-TG, 5-HT_4_-TG and WT with ergotamine. In brief, ergotamine (10 µM) increased left ventricular force of contraction and the rate-of-tension relaxation in the apex of hearts from H_2_-TG, 5-HT_4_-TG but not from WT (Table 1). Thus, the effects of ergotamine are not confined to the atrium but also present in the ventricle of the transgenic animals studied here.

### 2.4. Phosphorylation

In separate experiments, we tested the effects of ergotamine on the phosphorylation state of phospholamban in left auricular heart samples from H_2_-TG and 5-HT_4_-TG. These left atrial samples were electrically stimulated, and at the end of the concentration–response curves (10 µM ergotamine), the atria were frozen in liquid nitrogen. It turned out that ergotamine increased the phosphorylation state of phospholamban at the amino acid serine 16 in H_2_-TG and 5-HT_4_-TG compared to WT atrial preparations (Figure 6), suggesting a stimulation of cAMP-dependent protein kinases (Figure 1A).

### 2.5. Human Atrial Contraction

To find out whether our data are relevant in the human heart and thus clinically relevant, we performed the next contraction experiments in human cardiac preparations. In isolated electrically driven right atrial preparations, cilostamide, a phosphodiesterase III inhibitor, raised force of contraction to some extent. After this pre-stimulation, additionally applied ergotamine increased force of contraction further. This is exemplified in the original recording depicted in Figure 7. The mean data are put together in Figure 8 and depict this increase. Thereafter, the question arose whether the H_2_- or the 5-HT_4_-receptor mediated these effects. Therefore, we additionally applied first tropisetron to block 5-HT_4_- and thereafter cimetidine to block H_2_-receptors. As depicted in Figure 7 and summarized in Figure 8, the positive inotropic effect of ergotamine was not sensitive to tropisetron, but sensitive to additionally applied cimetidine. Based on these data, we would regard the positive inotropic effect of ergotamine in the human right atrium as H_2_-receptor-mediated.

## 3. Discussion

### 3.1. Main New Findings

The main new findings in this report as per the observation is that ergotamine can act as a functional agonist at both human 5-HT_4_- and human H_2_-receptors in the heart of appropriate transgenic mouse models. Importantly, ergotamine only uses H_2_-receptors to increase contractility in the human heart, specifically the human isolated atrium.

### 3.2. 5-HT_4_-Receptors

Looking at the chemical structure of ergotamine and knowing that it was synthesized from ergot constituents [9], knowing also that ergotamine can act on 5-HT_2A_-receptors in the periphery, we hypothesized that ergotamine might stimulate human cardiac serotonin receptors. In the human heart, serotonin only increases force via 5-HT_4_-receptors. Hence, we thought that ergotamine might stimulate 5-HT_4_ receptors in the human heart. As a first step, we used as a model our 5-HT_4_-TG [19]. Indeed, we noted a positive inotropic effect of ergotamine in the atrium and ventricle of 5-HT_4_-TG. In atrial preparations, we could show that ergotamine is less effective than serotonin and is thus functionally a partial agonist at 5-HT_4_-receptors because additionally applied serotonin raised the force further. The observations were different in right atrium, here, ergotamine was of similar efficacy as serotonin, and thus can be viewed as a full agonist. The reason for this discrepancy is unclear. One attractive hypothesis is that the overexpression of 5-HT_4_-receptors is higher in the sinus node than in the left atrium. However, this issue was beyond the scope of the present work.

### 3.3. H_2_-Receptors

It might be asked why we chose to study H_2_-TG. In other words, why should ergotamine stimulate cardiac histamine receptors? One could argue from a chemical point of view. If you look carefully at the structural formula of ergotamine (Figure 1B), you might discern an azole ring similar to the imidazole ring in histamine. In addition, we noted that ergometrine, a closely related lysergic acid derivative, can activate cardiac H_2_-receptors in guinea pig Langendorff-perfused hearts [14]. Moreover, LSD, or lysergic acid diethylamide, the well-known hallucinogenic drug also closely related to ergotamine, can stimulate rabbit H_2_-receptors and guinea pig H_2_-receptors in cardiac preparations [15]. Hence, we thought it worthwhile to test ergotamine in our H_2_-TG model system [23]. We noted that in H_2_-TG, as in guinea pig hearts and rabbit hearts [15], ergotamine acted as a partial functional agonist with regard to force of contraction. Moreover, as additionally applied histamine in atrial preparations from H_2_-TG hardly increased force or frequency above values reached by ergotamine itself, we would regard ergotamine as a fully functional agonist—with respect to force and beating rate at human H_2_-receptors expressed in the heart of H_2_-TG.

### 3.4. Mechanism of Ergotamine

Our assumption is that ergotamine acts as an agonist at cardiac human H_2_-histamine receptors because ergotamine increases force and beating rate only in atrium from H_2_-TG and not in WT. Likewise, we suggest ergotamine increases force and beating rate as an agonist at cardiac human 5-HT_4_-receptors because ergotamine only increases contractility in atrium from 5-HT_4_-TG and not in WT. Clearly, under our experimental conditions, ergotamine is a dual agonist at two receptors. One might hypothesize that the tryptamine ring of ergotamine binds to the binding pocket for tryptamine at the human 5-HT_4_-receptor. On the other hand, it can be claimed that the azole ring of ergotamine binds at the appropriate pocket of the human H_2_- receptors. There is sound evidence from crystallographic studies that ergotamine can bind with different functional groups to different receptors, but, in this case, to 5-HT_2A_-receptors and 5-HT_2B_-receptors [39]. Hence, there is precedence for our hypothesis.

Moreover, there are data on a functional interaction of H2-receptors and 5-HT4 receptors in human atrial tissue. Even the order of drug application was relevant. In more detail, when we first applied an increasing concentration of 5-HT, a positive inotropic effect followed. When we subsequently deposited histamine in increasing concentrations to the organ bath, the force declined. We speculated that this might mean that H2-receptors, context-dependently, can couple to first inhibitory and then stimulatory GTP-binding proteins, leading to a decrease and then increase in cAMP levels and therefore force of contraction [22]. One might speculate that such an effect might also be induced by ergotamine.

### 3.5. Role of Phosphorylation of Regulatory Proteins

Any sufficiently large H_2_-receptor stimulation or 5-HT_4_-receptor stimulation leads to an increase in the phosphorylation state of regulatory proteins that are substrates for the cAMP-dependent protein kinase [18,26]. Indeed, we described similar to others that histamine acting via H_2_-receptors in the heart can increase the phosphorylation state of phospholamban [40] in H_2_-TG and the human atrium [13]. Likewise, we and others reported that serotonin via 5-HT_4_-receptors can increase protein phosphorylation in 5-HT_4_-TG and human atrium [19,41,42]. For instance, serotonin increased the phosphorylation state of phospholamban in the isolated human atrial strips [42]. We extend here our previous studies by showing the ergotamine increase in H_2_-TG and in 5-HT_4_-TG the phosphorylation state of phospholamban. These phosphorylations can explain, at least in part, why ergotamine increased the relaxation rate in atrial and ventricular preparations from H_2_-TG and 5-HT_4_-TG.

### 3.6. Contractile Role of Ergotamine in Human Atrium

Interestingly, ergotamine acted only as an agonist at H_2_-agonist and not at 5-HT_4_-receptors in the isolated human atrium. One hypothetical explanation would be that the density of the overexpressed receptors is very high for 5-HT_4_-serotonin receptors in transgenic mouse atrium (5-HT_4_-TG), much higher than in the human heart. Usually, if the receptor is more highly expressed in a cell, less agonist is needed to stimulate this receptor. In other words, 5-HT_4_-TG is a very sensitive system to detect actions of drugs on human 5-HT_4_-receptors. Alternatively, if a drug does not stimulate 5-HT_4_-receptors in 5-HT_4_-TG, this drug is unlikely to increase human cardiac force via 5-HT_4_-receptors. We have mRNA data that indicate a thousand-fold overexpression of the 5-HT_4_ receptor in the heart from 5-HT_4_-TG [22]. However, these were non-failing mouse hearts and we do not know how the H2-receptor or 5-HT4 receptor expression might change in failing mouse hearts, as has been reported for failing rat hearts; in these failing rat hearts the mRNA for the 5-H4 receptor increased over time [43]. Indeed, binding is hardly detectible with radioligands of 5-HT_4_-receptors in human hearts, whereas expression of H_2_ in human hearts is easily measurable with ligand binding and is quite high [17,33]; thus, this is a plausible, albeit hypothetical, explanation for our data in the human atrium. Another explanation would be the partial agonistic effect of ergotamine on force of contraction in atrial preparations from 5-HT_4_-TG. In other words, the intrinsic activity of ergotamine for 5-HT_4_-receptors is lower than that of 5-HT. This second observation might also contribute to the missing effect of ergotamine in human atrium via 5-HT_4_-receptors. Hence, we tentatively conclude that in vivo effects of ergotamine in the human heart on force of contraction are more likely mediated by H_2_-histamine and not by 5-HT_4_–receptors.

### 3.7. Clinical Relevance

We would predict that a tachycardia after treatment with ergotamine in patients could be blocked by cimetidine, an approved drug. However, this prediction needs to be confirmed in a clinical study. Peak therapeutic plasma levels of ergotamine around 0.69 nM have been cited [44]. In intoxications in humans, much higher plasma levels of ergotamine, such as 0.015 µM, have been communicated [5]. Moreover, ergotamine is degraded by CYP2D6. Drugs that inhibit the activity of CYP2D6 could thus increase plasma levels of ergotamine. Indeed, some cases of ergotamine intoxication have been reported when patients were also given in addition to ergotamine other drugs that are inhibitors of CYP2D6 [45]. In addition, high ergotamine levels in plasma should occur in patients with a defective polymorphism of CYP2D6, because then less ergotamine will be degraded. In summary, there are various clinical situations where high plasma concentrations of ergotamine are reached. Under these high concentrations of ergotamine, H_2_-receptors in the heart might be stimulated by ergotamine. This stimulation of H_2_-receptors can lead to cardiac arrhythmias [13]. There are case reports that ergotamine taken in dosage to treat migraine can lead to an acute coronary syndrome and to arrhythmias in patients [46,47]. Based on our data, and with confidence, one could try cimetidine in such patients to terminate arrhythmias, namely atrial fibrillation.

### 3.8. Limitations of the Study

We have not tested the effects on the sinus node of man directly. Such a study would require access to the human pacemaker or corresponding stem cells. Such studies were beyond the scope of this initial study. We did not have the opportunity to study contractility in human left ventricular tissue for lack of access to that tissue. However, we argue that our studies in Langendorff-perfused hearts make it at least likely that ergotamine is also an agonist in the human left ventricle. Moreover, we cannot provide molecular information in which parts of the ergotamine molecule can interact with the H_2_-receptor or the 5-HT_4_-receptor. To this end, crystallographic studies would be required in subsequent work.

## 4. Materials and Methods

### 4.1. Transgenic Mice

The investigation conforms to the Guide for the Care and Use of Laboratory Animals published by the National Research Council (2011) [48]. Animals were maintained and handled according to approved protocols of the animal welfare committees of the University of Halle-Wittenberg, Germany. The generation and initial characterization of the transgenic mice were described before [19,23]. In brief, the human H_2_–histamine-receptor cDNA or the human 5-HT_4_-serotonin-receptor cDNA together with a C-terminal 6xhistidine tag were cut from the parent plasmid and inserted into the Eco ICR site of a mouse cardiac α-myosin heavy chain promoter expression cassette. For all experiments, 12–30 weeks-old transgenic mice and WT littermates of both sexes were used.

### 4.2. Contractile Studies in Mice

As reported often before, right or left atrial preparations from mice were isolated and mounted in organ baths (e.g., [20]). The buffer in the 10 mL-organ baths contained 119.8 mM NaCI, 5.4 mM KCI, 1.8 mM CaCl_2_, 1.05 mM MgCl_2_, 0.42 mM NaH_2_PO_4_, 22.6 mM NaHCO_3_, 0.05 mM Na_2_EDTA, 0.28 mM ascorbic acid and 5.05 mM glucose. The solution was continuously gassed with 95% O_2_ and 5% CO_2_ and maintained at 37 °C and pH 7.4 [20]. Spontaneously beating right atrial preparations from mice were used to study the intrinsic beating rate. The drug application was as follows. After equilibration was reached, ergotamine was cumulatively added to left atrial or right atrial preparations to establish concentration–response curves. Then, where indicated, either serotonin or histamine were additionally applied to the preparations (Figure 1B).

### 4.3. Contractile Studies on Human Preparations

The contractile studies on human preparations were done using the same setup and buffer as used in the mouse studies (see Section 2.2). The samples were obtained from sixteen patients. Fifteen patients were male and one patient was female. The mean age was 71 ± 10 years. Patient suffered from three vessel coronary heart disease and underwent bypass surgery. Drug therapy included metoprolol, furosemide, apixaban and acetyl salicylic acid. Our methods used for atrial contraction studies in human samples have been previously published and were not altered in this study [27].

### 4.4. Isolated Perfused Hearts

As described repeatedly from our group [19,23], isolated whole mouse hearts were retrogradely perfused with the same buffer as in Section 4.2 above. Hearts were allowed to beat by themselves. Force was monitored from the apex cordis by a hook connected to an electronic force monitor and digitized. Perfusion with drugs took place with a syringe connected to a pump. This pump was connected as a bypass to the aorta. At the end of the experiments, hearts were rapidly brought to the temperature of liquid nitrogen to stop any phosphorylation or dephosphorylation reactions. Frozen samples were kept at −80 °C until biochemical analysis.

### 4.5. Western Blotting

The homogenization of the samples, protein measurements, electrophoresis, primary and secondary antibody incubation and quantification were performed following our previously established protocols [23]. First antibodies were anti-calsequestrin (CSQ) antibody, Santa Cruz #sc390999 (diluted 1:20,000) and anti-phospholamban (pSer16) antibody, Badrilla A-010-12 (diluted 1:5000).

### 4.6. Data Analysis

Data shown are means ± standard error of the mean. Statistical significance was estimated using the analysis of variance followed by Bonferroni’s *t*-test or the Student’s *t*-test as appropriate. A *p*-value < 0.05 was considered to be significant.

### 4.7. Drugs and Materials

The ergotamine was in dissolved dimethylsulfoxide (DMSO), serotonin and histamine were dissolved in water and were purchased from Sigma-Aldrich (Germany). All other chemicals were of the highest purity grade commercially available. Deionized water was used throughout the experiments. Stock solutions were prepared fresh daily.

## 5. Conclusions

We can now answer the questions raised in the introduction in the following way: ergotamine increases force of contraction in both 5-HT_4_-TG and in H_2_-TG. In human atrial preparations, ergotamine increases force of contraction via the H_2_-receptors and not via 5-HT_4_-receptors.

## Figures and Tables

**Figure 1 ijms-24-04749-f001:**
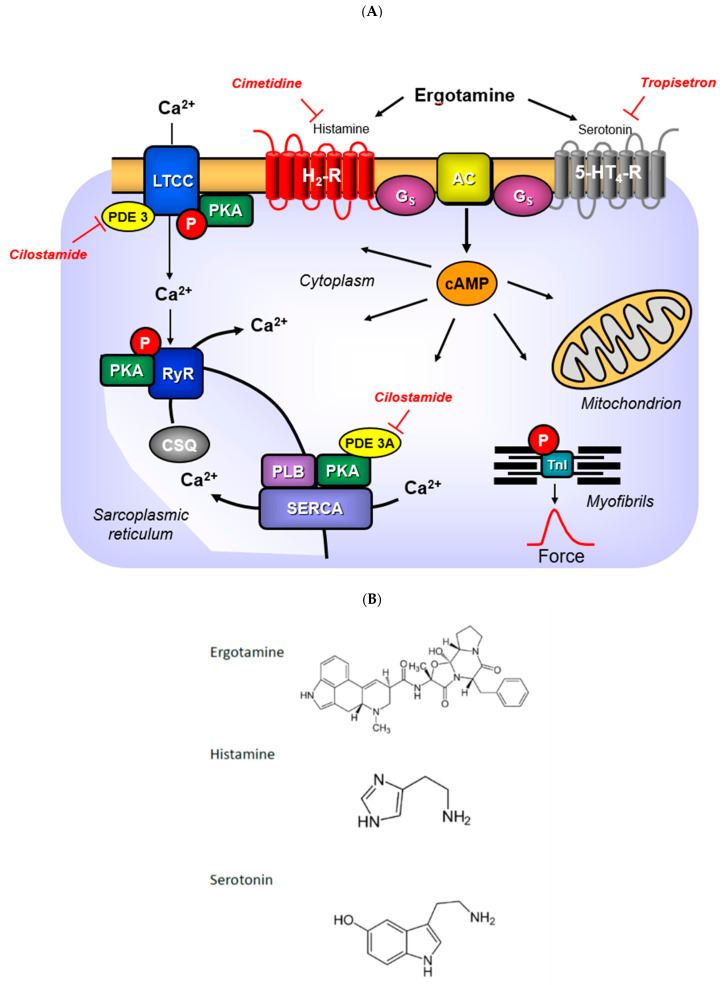
(**A**) Signal transduction: Serotonin activates and tropisetron antagonizes 5-HT4-receptors in the sarcolemma. Histamine activates and cimetidine antagonizes H2-receptors in the sarcolemma. Ergotamine can activate both receptors. Activation of these receptors lead via GTP binding stimulatory protein (Gs) an increased activity of adenylyl cyclases (AC). Thereby, more cAMP is generated. The formed cAMP is finally degraded by phosphodiesterases (PDE). The cAMP can activate cAMP-dependent kinases (PKA) that phosphorylate and activate target proteins in the heart. These included the L-type Ca channel (LTCC) that facilitates the entry of “trigger” Ca ion into the cell. Here, Ca ions can bind to myofilaments. Phosphorylation of the troponin inhibitor (TnI) or phospholamban (PLB) hasten cardiac relaxation. For instance, phosphorylation of PLB allows SERCA to pump more Ca ions from the cytosol into the sarcoplasmic reticulum. In the sarcoplasmic reticulum, Ca ions bind to calsequestrin (CSQ). The ryanodine receptor upon phosphorylation will allow more Ca ions to leave the sarcoplasmic reticulum and enter the cytosol. (**B**) Structural similarities: There are structural similarities between serotonin and ergotamine or histamine and ergotamine. Compare the azole ring in ergotamine to the imidazole ring in histamine. Compare the indole ring in ergotamine with the indole ring in serotonin. This may explain the activation of both receptors in (**A**) by ergotamine.

**Figure 2 ijms-24-04749-f002:**
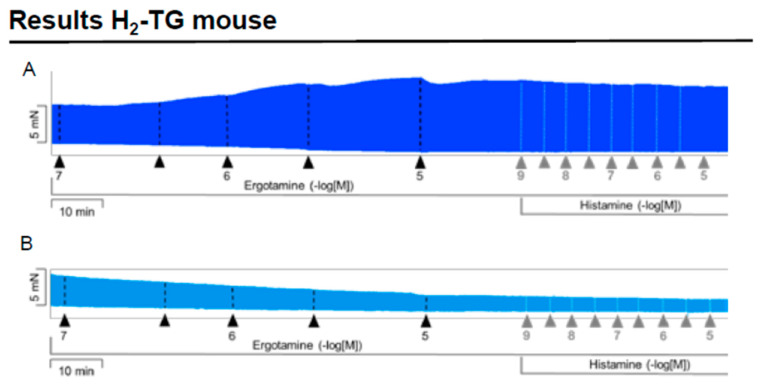
Ergotamine increases force in H_2_-TG. (**A**) Original recording in mouse left atrial preparations from H_2_-TG. It becomes apparent that ergotamine induced a time- and concentration-dependent positive inotropic effect. Black triangles indicate addition of ergotamine. Gray triangles indicate addition of histamine. (**B**) Original recordings in mouse left atrial preparations from WT. It becomes apparent that ergotamine did not induce a time- and concentration-dependent positive inotropic effect. (**C**) Summarized concentration–response curves for the effect of ergotamine or histamine on force of contraction. (**D**: T1) Time to peak tension. (**E**: T2) Time to relaxation. (**F**: dF/dt_max_) Rate-of-tension development. (**G**: dF/dt_min_) Rate-of-tension relaxation in H_2_-TG left atrial preparations. Ordinates in (**A**–**C**): force of contraction in milli-Newton (mN). Ordinate in (**D**) time to peak tension in milli seconds (ms). Ordinate in (**E**) time of relaxation in milli seconds (ms). Ordinate in (**F**) rate of contraction in mN/s and ordinate in (**G**) rate of relaxation in mN/s. Horizontal bars in (**A**,**B**) indicate time in minutes (min). Abscissae in (**D**–**F**) indicate concentrations of ergotamine or histamine in negative decadic logarithm of molar concentrations. * *p* < 0.05 vs. Ctr, first significant difference versus control (pre-drug value, Ctr). “n” indicates number of experiments given in brackets.

**Figure 3 ijms-24-04749-f003:**
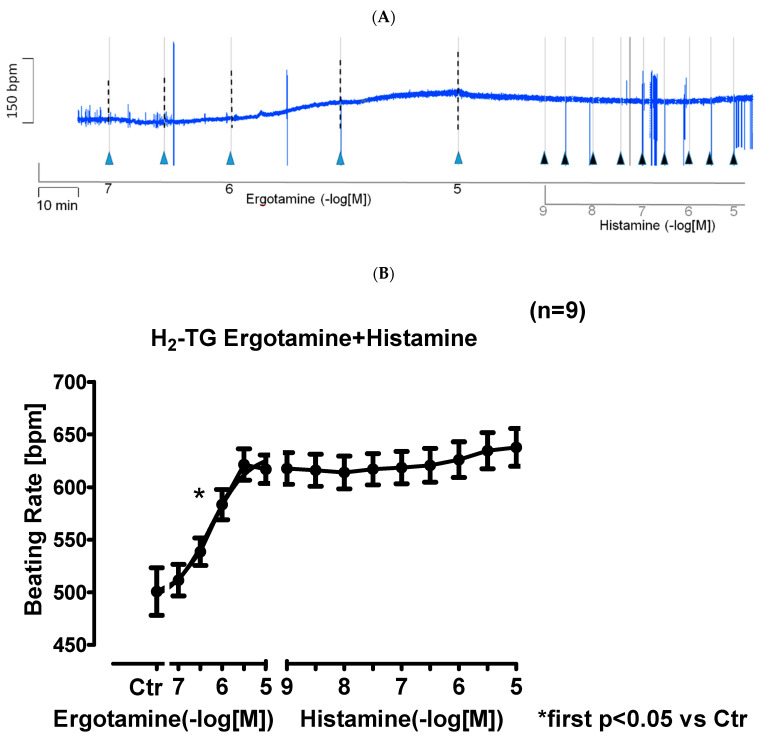
Ergotamine increases frequency in H_2_-TG: (**A**) Original recording of beating rate in H_2_-TG right atrium. Original recording: effect of ergotamine and histamine on beating rate in spontaneously beating right atrial preparations from H_2_-TG. The blue lines indicate duration of transient arrhythmias. Blue triangle mean addition of ergotamine and black triangles indicate addition of histamine. (**A**). Summarized effect of ergotamine and histamine on beating rate in spontaneously beating right atrial preparations from H_2_-TG (**B**). Ordinate in (**A**): beating rate in beats per minute (bpm). Ordinate in (**B**) in beating rate in beats pro minute (bpm). Abscissae in in (**A**,**B**), of ergotamine or histamine in negative decadic logarithms of molar concentrations. Horizontal bar in A indicates time axis in minutes (min). First significant differences versus control (Ctr; pre-drug value) are indicated by asterisks. “n” indicates number of experiments.

**Figure 4 ijms-24-04749-f004:**
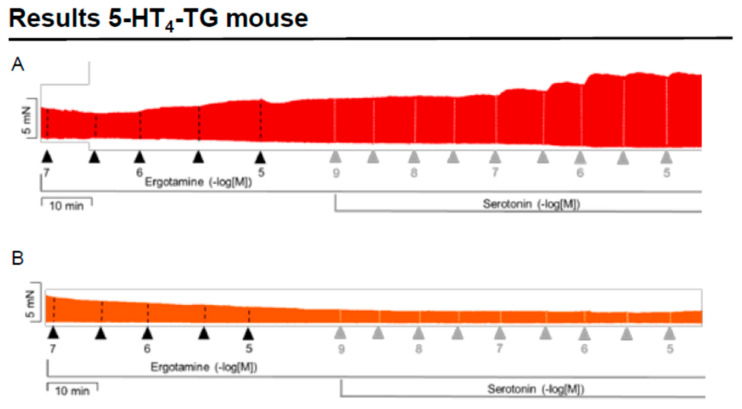
Ergotamine increases force in 5-HT_4_-TG. (**A**) Original recording in mouse left atrial preparations from 5-HT_4_-TG. It becomes apparent that ergotamine induced a time- and concentration-dependent positive inotropic effect. (**B**) Original recordings in mouse left atrial preparations from WT. It becomes apparent that ergotamine did not induce a time- and concentration-dependent positive inotropic effect. (**C**) Summarized concentration–response curves for the effect of ergotamine or histamine on force of contraction. (**D**: T1) Time to peak tension. (**E**: T2) Time to relaxation. (**F**: dF/dt_max_) Rate-of-tension development. (**G**: dF/dt_min_) Rate-of-tension relaxation in 5-HT_4_-TG left atria. * *p* < 0.05 vs. Ctr, first significant difference versus control (pre-drug value, Ctr). Ordinates in (**A**–**C**): force of contraction in milli-Newton (mN). Ordinate in (**D**) time to peak tension in milli seconds (ms). Ordinate in (**E**) time of relaxation in milli seconds (ms). Ordinate in (**F**) rate of contraction in mN/s and ordinate in (**G**) rate of relaxation in mN/s. Horizontal bars in (**A**,**B**) indicate time in minutes (min). Abscissae in (**D**–**F**) indicate concentrations of ergotamine or histamine in negative decadic molar concentrations. Significant differences versus control (CTR; pre-drug value) is indicated in asterisks. “n” indicates number of experiments.

**Figure 5 ijms-24-04749-f005:**
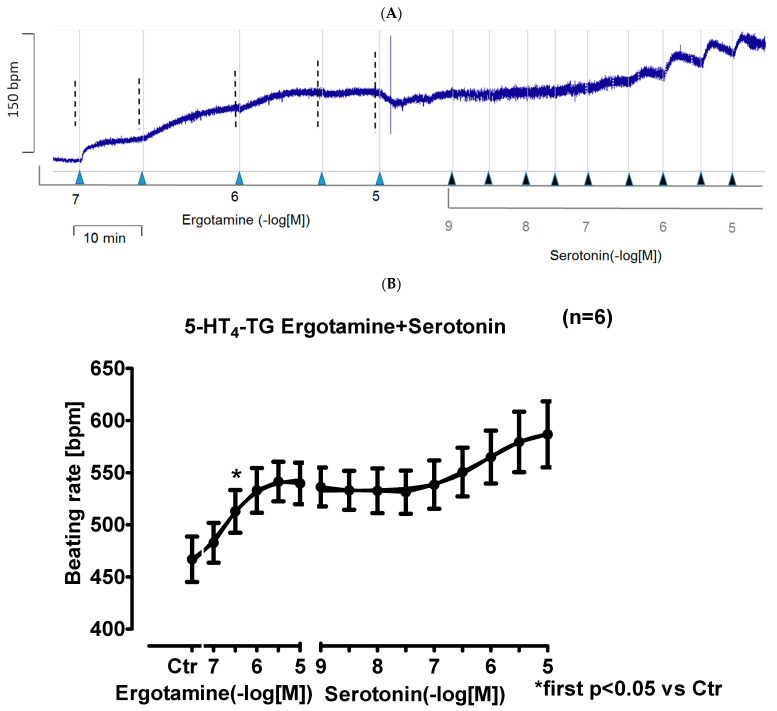
Ergotamine increases frequency in 5-HT_4_-TG. (**A**) Original recording: effect of ergotamine and serotonin on beating rate in spontaneously beating right atrial preparations from 5-HT_4_-TG. (**B**): Summarized effect of ergotamine or serotonin on beating rate in spontaneously beating right atrial preparations from 5-HT_4_-TG. Abscissae in (**B**) give concentrations of ergotamine or histamine in negative decadic logarithms, respectively. Horizontal bar in (**A**) indicates time axis in minutes (min). In (**B**) first significant differences versus control (Ctr; pre-drug value) is indicated in asterisks. “n” indicates number of experiments.

**Figure 6 ijms-24-04749-f006:**
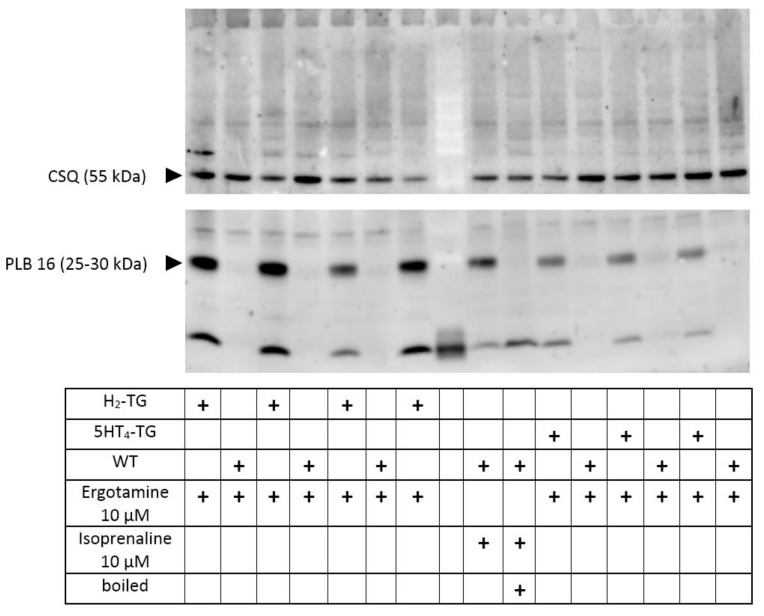
Ergotamine increases the phosphorylation state of phospholamban: contracting preparations as seen in Figure 2 and Figure 4 were freeze-clamped after the addition of ergotamine 10 µM (see fourth row in the table below the Western blot). This was done with left atrial preparations from wild type mice (WT) as negative control, and also in atrial preparations from 5-HT_4_-TG or H_2_-TG as indicated by “+”. No additional histamine or serotonin was added. Typical Western blots are seen. Western blots depict serine 16 phosphorylated phospholamban (PLB) with arrows. As a loading control, we assessed the protein expression of calsequestrin (CSQ) by cutting the lanes of the blot horizontally and incubating the lower and upper halves with different primary antibodies. Relevant molecular weight markers are indicated with arrows and marked with kilodalton (kDa). As a positive control, samples from atrium treated with 10 µM isoprenaline are shown in the middle. Boiled means that the isoprenaline-treated atrium was brought to the temperature of 95 °C for 10 min and was then rapidly loaded on this original gel. Please note the mobility shift in the boiled sample confirming that we really detected phospholamban. In the middle row (empty), we allowed the molecular weight marker to run.

**Figure 7 ijms-24-04749-f007:**
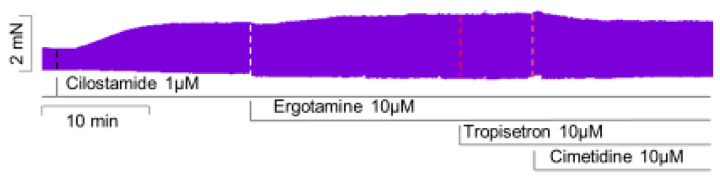
Ergotamine increases force in human atrium: original recording of the effect of 10 µM ergotamine (Ergo), 10 µM tropisetron (Tropi) and 10 µM cimetidine (Cime) in the presence of 1 µM cilostamide (Cilo) on HAP.

**Figure 8 ijms-24-04749-f008:**
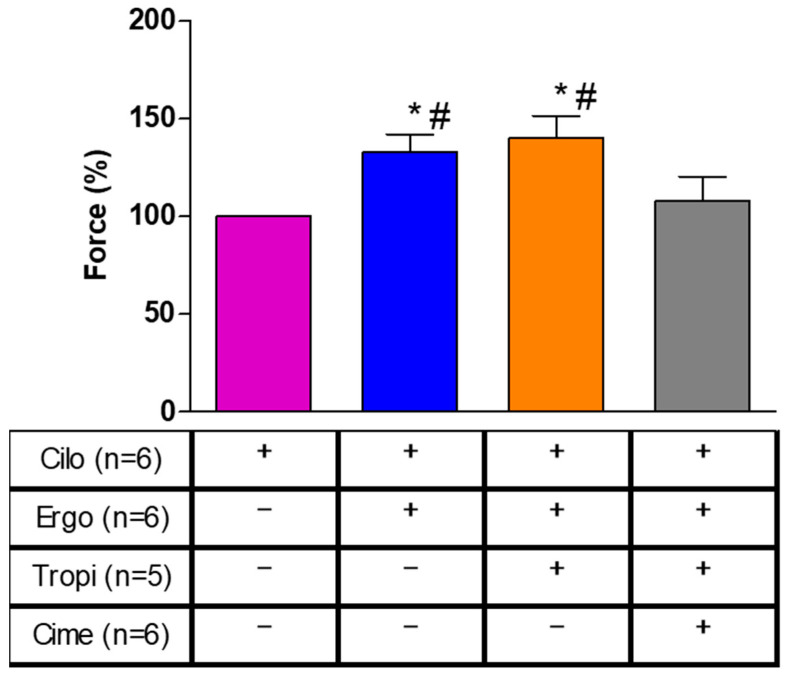
Effects of subsequently applied cilostamide (Cilo, 1 µM), then additionally ergotamine (10 µM), tropisetron (Tropi, 10 µM) and cimetidine (Cime, 10 µM) on force of contraction in isolated electrically stimulated human right atrial preparations. Compare with Figure 7 for the order of drug application. Numbers in brackets indicate the number of muscle strips studied. Force of contraction was calculated by setting the maximum effect of cilostamide as one hundred percent (100% compare Figure 7). * or # indicate significant differences (*p* < 0.05) against Cilo or Cime, respectively.

**Table 1 ijms-24-04749-t001:** Maximum effect of ergotamine (10 µM) on force of contraction in milli-Newton (mN) and the rate-of-tension relaxation mN/seconds (mN/s) in isolated perfused hearts from H_2_-TG, 5-HT_4_-TG and WT. # indicate *p* < 0.05 versus pre-drug (ergotamine) value. N = number of animals.

	WT	H_2_-TG	5-HT_4_-TG
N=	7	7	4
Basal force (mN)	11.3 ± 1.2	11.3 ± 1.5	14.9 ± 2.1
Force after ergotamine (mN)	12.3 ± 1.2	15.8 ± 1.8 #	18.5 ± 2.9 #
Basal rate of relaxation (mN/s)	224 ± 14.1	208 ± 28.4	218 ± 14.0
Rate of relaxation after ergotamine (mN/s)	250 ± 18.6	315 ± 44.8 #	287 ± 27.8 #

## Data Availability

The data of this study are available from the corresponding author upon reasonable request.

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
