# Peer review of "Ergotamine Stimulates Human 5-HT4-Serotonin Receptors and Human H2-Histamine Receptors in the Heart"

_ijms, 2023, doi:10.3390/ijms24054749_

Round 1

Reviewer 1 Report

Dear colleagues,
In this manuscript, the authors study influence of  Ergotamine  on human 5-HT4-serotonin receptors and 2 human H2-histamine receptors in the heart. The results are very interesting. The figures reflect the results of the study. Despite the very good impression of the article, there are some remarks which are insignificant but could improve the article in my opinion, partly:

There are some stylistic and grammar mistakes. For example “… only as an agonist on H2-histamine…” in introduction. Also, we think it is necessary to replace “=” for something like “which are…” in “in H2-TG (=mouse with…” and “…from 5-HT4-TG (=mouse with…”

Obtaining of human preparations is unclear and needs more detail description in M&M despite reminding of sources for it.

Article is finished with Summary which is better to replace for Conclusion and change sense according to that.

Some self-citations could be rejected.

In summary, I have been satisfied with the high level of the article. I believe this manuscript will attract significant attention from the research community. In my personal opinion, the article is very valuable, a great prospect for further research, and can be recommended for publication.

Reviewer 2 Report

I read with interest the manuscript entitled: "Ergotamine stimulates human 5-HT4-serotonin receptors and 2 human H2-histamine receptors in the heart.

The following are minor comments addressed to the authors:

The work carried out by the authors is interesting, although very useful information is missing from this work, namely the effects of ergotamine in human left ventricular tissue. The mechanistic data on the interaction of ergotamine with the H2 receptor or the 5- HT4 receptor. The authors , however the authors did mention this in the manuscript.

it would be informative to mention.

- In the Clinical relevance section:

it would be informative to mention that it has been reported vasospastic effects of low-dose ergotamine in the treatment of migraine. (Exp Clin Cardiol. 2012 Spring; 17(1): 43–44.)

- Could the authors go further into  the pharmacotherapeutic implication of the dual agonist effect on 5-HT4 receptors and H2 receptors

- It would be interesting to see the ratio of the relative density of the two receptors (5-HT4 and H2) in the situation of heart failure knowing that there is an increase of 5-HT4 mRNA in heart failure (Brattelid, T.et al Naunyn-Schmiedeberg's Arch. Pharmacol. 2004, 370, 157-166)
